

# Exploring a model-based analysis of patient derived xenograft studies in oncology drug development

Jake Dickinson[1,*], Marcel de Matas[1], Paul A. Dickinson[1] and Hitesh B. Mistry[1,2,*]

[1] Seda Pharma Development Services Ltd., Alderley Edge, United Kingdom
[2] Division of Pharmacy, University of Manchester, Manchester, United Kingdom
[*] These authors contributed equally to this work.

## ABSTRACT

**Purpose**. To assess whether a model-based analysis increased statistical power over an analysis of final day volumes and provide insights into more efficient patient derived xenograft (PDX) study designs.

**Methods**. Tumour xenograft time-series data was extracted from a public PDX drug treatment database. For all 2-arm studies the percent tumour growth inhibition (TGI) at day 14, 21 and 28 was calculated. Treatment effect was analysed using an un-paired, two-tailed $t$-test (empirical) and a model-based analysis, likelihood ratio-test (LRT). In addition, a simulation study was performed to assess the difference in power between the two data-analysis approaches for PDX or standard cell-line derived xenografts (CDX).

**Results**. The model-based analysis had greater statistical power than the empirical approach within the PDX data-set. The model-based approach was able to detect TGI values as low as 25% whereas the empirical approach required at least 50% TGI. The simulation study confirmed the findings and highlighted that CDX studies require fewer animals than PDX studies which show the equivalent level of TGI.

**Conclusions**. The study conducted adds to the growing literature which has shown that a model-based analysis of xenograft data improves statistical power over the common empirical approach. The analysis conducted showed that a model-based approach, based on the first mathematical model of tumour growth, was able to detect smaller size of effect compared to the empirical approach which is common of such studies. A model-based analysis should allow studies to reduce animal use and experiment length providing effective insights into compound anti-tumour activity.

Corresponding author
Jake Dickinson,
jake.dickinson@sedapds.com

## INTRODUCTION

Preclinical Oncology drug development is heavily reliant on xenograft studies to assess the anti-tumour effect of new compounds (*Jung, 2014*). These studies can vary in duration, as short as 14 days (*Hather et al., 2014*), to 4 weeks (*Xu et al., 2018*), and even longer depending on the control growth rate. They represent the first opportunity, during development, to assess how the kinetics of drug disposition affects the kinetics of tumour growth

(*Plowman et al., 1997*). The xenograft study starts by grafting a human cell-culture into the flank of an immunocompromised mouse. Digital callipers are then used to measure the length and width or the length, width and height to calculate tumour volume at regular time intervals. Once the grafted tumour has reached a certain pre-specified volume each xenograft is randomised into one of the treatment arms or the untreated arm (control arm within the study). Thus, across all study arms the volume at randomisation is comparable. The treatment effect is then calculated by measuring the difference in mean tumour volumes between the treated and control group at the end of the study (*Corbett et al., 2004*). This metric is typically referred to as the Tumour Growth Inhibition (TGI) value and is calculated as follows:

$$\%\text{TGI}(T=t) = \frac{V(\text{control}, T=t) - V(\text{treated}, T=t)}{V(\text{control}, T=t) - V(\text{control}, T=0)}$$

where $V(\textit{treated}, T=t)$ and $V(\textit{control}, T=t)$ are the mean volumes of the treated and control group at time $T=t$, the end of the study and $V(\textit{control}, T=0)$ is the mean volume of the control group at the time of randomisation. To assess whether the volumes at time $t=T$ are significantly different between the control and treated arms of the study an un-paired, two-tailed $t$-test is performed and the resultant $p$-value is usually reported together with the TGI value.

The above approach to analysis of TGI data clearly ignores the time-series that is generated up until the TGI value is recorded. Furthermore, the TGI value can become biased if mice have dropped out at a time-point before the TGI value is calculated. This is common in the control arm due to the volume exceeding a pre-defined animal welfare limit. Thus, performing an un-paired, two-tailed $t$-test on the final day of the study, results in an under-prediction of the mean control volume and hence the efficacy of the treatment is underestimated. This issue, however, can be resolved by calculating TGI at an earlier time-point where no drop-outs exist or a joint longitudinal drop-out model could also be used (*Martin, Aarons & Yates, 2016*).

An alternative to performing an un-paired, two-tailed $t$-test on the final study day is to perform a model-based regression analysis. The key advantage of performing a regression analysis over the approach discussed is that all data points in the time series are used. This will lead to an increase in statistical power, hence reduce the number of animals used and thus reduce the cost of a xenograft study.

A previous study, by *Hather et al. (2014)*, has shown that a model-based regression approach is likely to improve the power of xenograft studies over doing an un-paired, two-tailed $t$-test. However, the Hather et al. study did not consider the application of such an approach to patient derived xenografts (PDX), which are known to have higher variance than the standard cell-line xenograft and are becoming more popular within preclinical development. Furthermore, the approach by Hather et al. did not consider the use of mixed-effects/hierarchical modelling approach which may likely further increase the statistical power of such studies.

In this study we build on the work by Hather et al. by assessing the increase in power obtained by using a model-based mixed-effects regression analysis and a naïve pooled
approach over the typical un-paired, two-tailed $t$-test analysis of final volumes for a large open PDX database (*Gao et al., 2015*). In addition, we also analysed a traditional standard cell-line xenograft study (*Knutson et al., 2016a*) for comparison to the patient-derived xenograft analysis and perform a brief simulation study highlighting the merits of a model-based mixed-effects regression analysis.

There are numerous mathematical models that can be used within a model-based analysis of tumour growth data (*Ribba et al., 2014*). Within this study we have chosen to use the first tumour growth model developed by Mayneord in 1932 (*Mistry, Orrell & Eftimie, 2018*). This model has been derived from first principles and recently been shown to provide a good description of both preclinical and clinical tumour size time-series data (*Mistry, Orrell & Eftimie, 2018*; *Orrell & Mistry, 2019*).

## METHODS

### Real data study
#### *Xenograft data*
Data from a study by *Gao et al. (2015)* which analysed 1000 PDX models across 59 treatments and controls was collected. The data was then grouped creating 59 two arm studies involving a control and treatment arm (hence a 'study' refers to a different treatment where, within that study, there was 2 arms, a treatment and control arm). This data was then truncated to either 14, 21 or 28 days, to mimic typical lengths of xenograft studies, for analysis.

#### *Empirical approach: un-paired, two-tailed T-test*
On the final day of the study, day 14, 21 or 28, an un-paired, two-tailed $t$-test between the control and treated volumes was conducted with a $p$-value and %TGI reported.

#### *Model-based approach: likelihood ratio-test*
The time-series from the animals used within the empirical approach were used for the following model-based analysis. By assuming that the tumour is spherical we first converted tumour volume to radius using:

$$R = \left(\frac{3V}{4\pi}\right)^{\frac{1}{3}}$$

where $V$ is the volume of the tumour and R the radius.

Next, we fitted the following model, whose mechanistic derivation can be found in *Mistry, Orrell & Eftimie (2018)*, *Orrell & Mistry (2019)* and *Mayneord (1932)* and in the Supplemental Information, to the combined two arm data-set:

$$
\begin{aligned}
R_{ij} &= \left(a_i + b_i t_{ij}\right)\left(1 + e_{ij}\right) \\
\log(a_i) &\sim N\left(\mu_1, \sigma_1^2\right) \\
\log(b_i) &\sim N\left(\mu_2, \sigma_2^2\right) \\
e_{ij} &\sim N\left(0, \sigma_3^2\right).
\end{aligned}
\tag{1}
$$
where $R_{ij}$ is the observed radius of xenograft $i$ at time $j$, $a_i$ is the value of the radius at time 0, time of randomisation, for xenograft $i$, $b_i$ is the rate of growth of the radius for xenograft $i$, $t_{ij}$ is the time-point for xenograft $i$ at time $j$ at which the observation was recorded and $e_{ij}$ is the residual error for xenograft $i$ at time $j$. Note, that a proportional error model was used as the variability in tumour size grew over time i.e., we have heteroscedastic variance (see Supplemental Information for more details). We then modified the distribution of $b_i$ and introduced a population treatment effect parameter $c$ to account for difference between control and treated growth rates in the following way

$$b_i \sim N\left(\mu_2 + cT_i, \sigma_2^2\right)$$
$$T_i = \begin{cases} 0 \text{ if control} \\ 1 \text{ if treated} \end{cases} \tag{2}$$

and re-fitted the model to the data. Since the models considered are nested the likelihood ratio-test (LRT) was used to assess if adding a treatment effect improved model fit to the data over a model with no treatment effect.

In addition to regressing against radius an analysis regressing against volume was also considered. The model used for the volume analysis was as follows:

$$V_{ij} = \frac{4\pi \left(a_i + b_i t_{ij}\right)^3}{3} \left(1 + e_{ij}\right)$$
$$\log(a_i) \sim N\left(\mu_1, \sigma_1^2\right)$$
$$\log(b_i) \sim N\left(\mu_2, \sigma_2^2\right)$$
$$e_{ij} \sim N\left(0, \sigma_3^2\right). \tag{3}$$

where $V_{ij}$ is the observed radius of xenograft $i$ at time $j$. The assessment of treatment effect was conducted in a similar way as stated above.

In addition to using mixed-effects we also conducted a naïve pooled data modelling approach to assess treatment effect i.e., using just fixed-effects. This analysis was conducted using the radius model only.

All parameter estimation for the mixed-effect models was conducted using the *saemix* library in R. Parameter estimation for the pooled data approach was done using the *minpack.LM* library in R.

## Simulated data study: power calculation

A 14, 21 and 28 day 2-arm simulation study, using model (1) stated above, was conducted in the following way. All parameter values except for the treatment effect $c$ were based on fitting model (1) to all control data up to day 14, 21 or 28. The treatment effect parameter $c$ was chosen to give either a 50% or 100% TGI at 14, 21 or 28 days. The sample size for each arm of the simulated study was set at 5, 8, 10, 12 and 15. When simulating the tumour volumes, we assumed that the measurements were taken on the same days as in the Gao et al. study, so every 3 to 4 days. In addition to using the Gao et al. study to

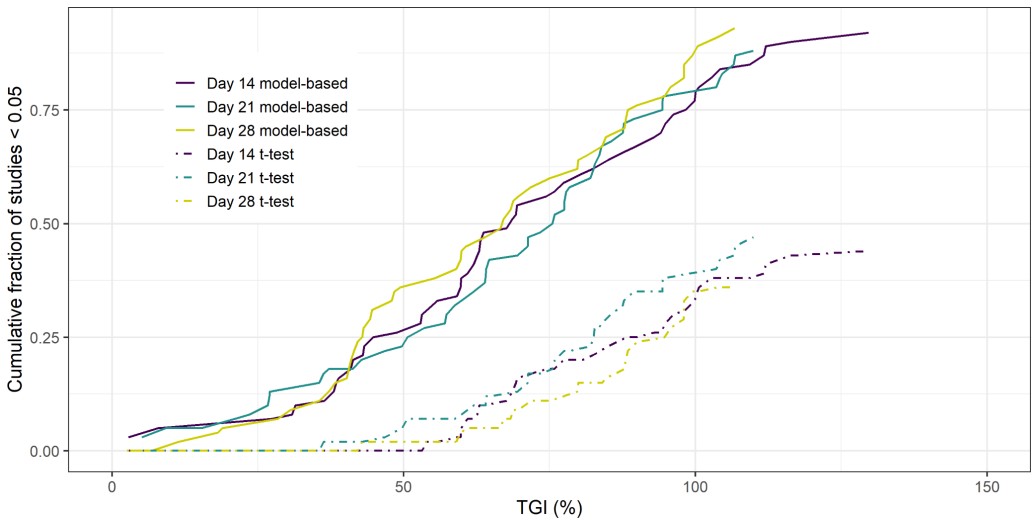

**Figure 1** Cumulative fraction of studies $p < 0.05$ vs TGI.

generate model simulation parameters we also utilised a standard cell-line (CDX) study by Knutson et al., which investigated the effect of paclitaxel and paclitaxel conjugated to $\alpha$-CEA-680-PTX antibody in a BxPC-3 model cell line (*Knutson et al., 2016a*), data was taken from datadryad.org (*Knutson et al., 2016b*).

For each simulated study, control and treatment arm, the empirical and model-based approaches were applied with $p$-values recorded. This was done 1000 times for a given simulation setup with the proportion of $p$-values <0.05 recorded and visualised.

## RESULTS

### Real data study

The results of the 14, 21 and 28 day 2-arm, treatment versus control, study analysis of the PDX data can be seen in Fig. 1. The Figure shows the cumulative fraction of $p < 0.05$ studies as a function of TGI for both the model-based and empirical approach. The key result is that regardless of the duration of the study the model-based approach gives greater statistical power over the empirical approach. Furthermore, this increase in power leads to the model-based approach being able to detect smaller TGI values (25%) compared to the empirical approach (50%).

Similar results were seen when regressing against volume rather than radius (see Supplemental Information). If the analysis was done using a naïve pooled approach for radius, a reduction in power was seen when the study endpoint was 14 days compared to the mixed-effects model. However, this difference was no longer apparent for longer studies. (see Supplemental Information).

### Simulated data study

The results of the simulation study using parameters derived from analysing the controls of the PDX study can be seen in Table 1. It shows that the power of the model-based approach

**Table 1  Results of the simulated PDX power analysis from the empirical and model-based approach.**

| | | Percentage of p-values <0.05 | | | | | |
|---|---|---|---|---|---|---|---|
| TGI (%) | N | Empirical Day 14 | Model-Based Day 14 | Empirical Day 21 | Model-Based Day 21 | Empirical Day 28 | Model-Based Day 28 |
| 100 | 15 | 61 | 71.1 | 65.3 | 85.6 | 68.6 | 100 |
| | 12 | 50.3 | 70 | 53.5 | 85.5 | 53.9 | 100 |
| | 10 | 48.8 | 70.3 | 47.4 | 83.1 | 45.8 | 99.7 |
| | 8 | 42.6 | 69.4 | 40.1 | 83.3 | 39 | 99.7 |
| | 5 | 25.8 | 66 | 23.5 | 78 | 17 | 98.5 |
| 50 | 15 | 16 | 54.4 | 15.2 | 77.7 | 6.7 | 99.2 |
| | 12 | 10.6 | 46.9 | 10.7 | 71.9 | 5 | 96.9 |
| | 10 | 10 | 47.9 | 10.7 | 63.5 | 3.9 | 95 |
| | 8 | 8.1 | 43.7 | 8.8 | 59.4 | 3.5 | 93.7 |
| | 5 | 3.1 | 37.1 | 2.8 | 51.7 | 1.7 | 78.1 |

Notes.
    TGI, tumour growth inhibition.

**Table 2  Comparison of PDX and CDX power analyses.**

| | | | Percentage of p-values <0.05 | | | |
|---|---|---|---|---|---|---|
| | | | Empirical | | Model-Based | |
| TGI (%) | N | Day | CDX | PDX | CDX | PDX |
| 100 | | 14 | 71.3 | 48.8 | 71.4 | 70.3 |
| 50 | | | 39.5 | 10 | 54 | 47.9 |
| 100 | 10 | 21 | 85.2 | 47.4 | 85.7 | 83.1 |
| 50 | | | 48.1 | 10.7 | 73.2 | 63.6 |
| 100 | | 28 | 99.6 | 45.8 | 100 | 99.7 |
| 50 | | | 50.3 | 3.9 | 98.1 | 95 |

Notes.
    TGI, tumour growth inhibition; CV, coefficient of variation; CDX, cell-line derived xenograft; PDX, patient derived xenograft.

is greater than that using the empirical approach. We can see that the power does decrease as we decrease the number of mice, as expected. A further simulation power analysis was done using parameters derived from a CDX study, details on the parameter values can be found in Supplemental Information. The result of this analysis in comparison to the PDX analysis can be seen in Table 2. The results show that if the variability is low and a 100% TGI is sought, then the power of using an empirical versus model-based approach is similar. However, in the other scenarios explored we found that the model-based approach has greater power over the empirical approach. The table further highlights the difference in power between CDX and PDX studies; this is due to the increased variability in time-series seen in PDXs, see Supplemental Information for parameter estimates.

## DISCUSSION

It is well known that cell-line derived xenografts (CDX) that have been established over the decades poorly mimic human disease (*Williams, 2018*). Thus, there has been a long-standing

interest in developing new animal models that better mimic patient tumours and their microenvironments. One approach that has been gaining favour in recent years is the use of PDXs. These models show more variability in their time-series and treatment response than their cell-line derived counterparts mainly due to the increased heterogeneity within the sample used (*Day, Merlino & Van Dyke, 2015*). Given the increased cost of using PDXs versus CDXs more importance should be placed on how these studies are analysed than is presently done.

In this study we explored the typical empirical based analysis methods using the final volumes to model-based approaches that use the time-series up to a chosen endpoint across 59 2 arm trials taken from a publicly available PDX database (*Gao et al., 2015*). The empirical approach consisted of applying an un-paired, two-tailed $t$-test to the final volumes of the control versus treated arms of a study. The model-based approach, however, involved using a parametric model to describe the time-series and the LRT to assess if including a treatment effect parameter improved model fit. Thus, the model-based approach used all the data whereas the empirical based approach did not. Regarding the model-based approach we explored 3 options: (1) using radius within a mixed-effects framework; (2) using volume within a mixed-effects framework and (3) using radius within a naïve pooled approach. It must be noted that in many xenograft experiments tumour volume is not measured directly but calculated from two length measurements. Errors in measurement are therefore made on the length scale not on the volume scale. Thus, regressing against a length may be more appropriate.

The results showed that the model-based approaches had more statistical power than the empirical based approach. Regarding the type of modelling approach, a modest difference was seen between the mixed-effects approach and the naïve pooled approach and no difference seen between regressing against volume versus radius. The key result is consistent with larger studies involving CDXs that have been previously reported (*Hather et al., 2014*). We found that the model-based approach could identify TGI values as low as 25% whereas >50% TGI is required for the empirical based method to detect a difference. Given that we would expect modest TGI with PDXs compared to CDXs, due to increased heterogeneity (*Gao et al., 2015*), this highlights the importance of using a model-based approach for such analyses.

A second key result was that due to lower variability in controls between CDXs compared to PDXs, no difference in statistical power between the model-based and empirical based approach was found when TGI was 100%. This observation supports the common practice of using the empirical based approach to analyse CDX data when high TGI are expected. This was not the case when moving to PDXs, where an increase in power was observed using a model-based over empirical based approach indicating that model-based approaches should become the common practice for analysing PDX data.

In summary, the results have shown that detecting modest TGI values such as 50% a model-based approach will lead to increased power for both CDX and PDX studies. *Wong et al. (2012)* have shown that clinical exposure targets that relate to at least 60% TGI in CDX studies are required to see clinical efficacy. Thus, based on the power analysis

conducted here a model-based analysis would be the most appropriate approach to detect such modest efficacy in both PDX and CDX studies.

Unlike previous studies conducted within this field we chose to use the same endpoint, percent TGI, when comparing analysis methods. It must be noted that the model-based approach does not calculate TGI directly from the data but the rate of change in tumour size. If a model -based analysis is performed one option is to report a model estimated TGI, which can be calculated via simulation. This is an important factor to highlight as by performing this simulation, as part of the model-based analysis, the experimental community can continue to use this familiar measure of efficacy which we hope will encourage the community to explore model-based xenograft analysis approaches as we enter the age of more sophisticated animal models in cancer.

### Funding
The authors received no funding for this work.

### Competing Interests
Marcel de Matas and Paul A. Dickinson are Directors and Shareholders; Jake Dickinson is an employee and Hitesh B. Mistry is a contractor of Seda Pharmaceutical Development Services Ltd. which provides modelling services to the Pharmaceutical Industry.

### Author Contributions
- Jake Dickinson and Hitesh B. Mistry conceived and designed the experiments, performed the experiments, analyzed the data, prepared figures and/or tables, authored or reviewed drafts of the paper, and approved the final draft.
- Marcel de Matas and Paul A. Dickinson analyzed the data, authored or reviewed drafts of the paper, and approved the final draft.

### Data Availability
R scripts and raw TGI measurements are available in the Supplemental Files.

The datasets analysed during the current study are available in Gao et al. (2015)'s Table S1.

### Supplemental Information
Supplemental information for this article can be found online at http://dx.doi.org/10.7717/peerj.10681#supplemental-information.

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
