# Peer review of "Exploring a model-based analysis of patient derived xenograft studies in oncology drug development"

_PeerJ, doi:10.7717/peerj.10681_

## Round 0.1 · original submission · Minor Revisions

Your manuscript has now been reviewed and the reviewers' comments are appended below. I agree with them in finding overall your manuscript relevant, interesting, and well written. However, they have raised points that need to be addressed by a minor revision. We, therefore, invite you to revise and resubmit your manuscript, taking into account the points raised and carefully answering them. Please submit a revised paper and a point-by-point response to the referees. We look forward to hearing from you soon.
Best

Reviewer 1 ·

Basic reporting

The manuscript is well written with sufficient literature references. It builds upon previous reports of model based approaches to interpreting xenografted mouse cancer model data with the novelty here of the application to patient derived xenografts that are significantly more heterogenous in their response compared to the cell line derived xenografts that have been cultured in vitro. Some small queries/comments:
1. Are "studies" referring to the different treatments tested in Gao et al?
2. I suggest adding references to back up the claims of the applicability of the Mayneord model on line 77
3. Is figure 3 of the control growth only?
4. Please report the treatment effect parameter c where this was found to be significant. This could be added to table 5.

Experimental design

The data are reported in the literature and so the authors had no influence on the design of the in vivo experiments. The analysis approach, including the application of nonlinear mixed-effects, is appropriate. I have no further comments.

Validity of the findings

The findings are supported by the data shown.

Additional comments

The manuscript reports an analysis approach that adds significant value to the interpretation of animal data. The approaches reported can be implemented in software available to the biostatistics community and so provide a practical improvement on current methodology.

Reviewer 2 ·

Basic reporting

no comment

Experimental design

no comment

Validity of the findings

no comment

Additional comments

Dickinson et al. describe a study designed to provide an analysis of the approaches used to assess the outcome of animal xenograft studies used in preclinical drug discovery.
Critically, the authors find that empirical (end-point) analysis is inferior to the model based analysis and this has conseuqences to the length and size of subsequent in-vivo studies. This finding isn’t entirely new, and the authors acknowledge the work previously by Hather et al, but most critically extends from the Hather study by assessing the applicability to patient-derived xenograft (PDX) as opposed to cell-derived xenograft (CDX). This is important because PDX studies are potentially more translatable to human studies due to the source material for xenografting, and consequently more costly to run and have greater variance in measurements. The consequences for such work are manifold, greater efficiency in the collection of data, reduction in number of animals and greater precision (and confidence) in the study conclusions, that in turn has onward implication to drug progression to clinical phase trials.

The work is well written and approachable, below are some minor comments and questions to be considered:

• The authors describe using one of many such tumour growth models (Mayneord, 1932). It would be interesting to analyse (or comment) on the expected effects of different tumour growth models on the results.
• Given the model-based approach considers the entirety of data generated up until the study cut off, i.e. 14, 21 and 28 day effects. Could the approach be used to cut short any future xenograft studies once significance, using model based approaches has been found?
• The figures appear to be quite low-resolution, can high resolution images be included for final publication.

---

## Round 0.2 · accepted · Accept

Your manuscript has now been reviewed again by one of our reviewers. In light of their advice, I am delighted to say that we are happy to publish this suitably revised version of your manuscript in PeerJ.

Reviewer 1 ·

Basic reporting

The manuscript is well written. The authors have answered all of my comments

Experimental design

No comment

Validity of the findings

No comment

Additional comments

Thank you for adressing my comments